# Why are Spanish Adolescents Binge Drinkers? Focus Group with Adolescents and Parents

**DOI:** 10.3390/ijerph17103551

**Published:** 2020-05-19

**Authors:** José Manuel Martínez-Montilla, Liesbeth Mercken, Marta Lima-Serrano, Hein de Vries, Joaquín S. Lima-Rodríguez

**Affiliations:** 1Department of Health Promotion, School for Public Health and Primary Care CAPHRI, Maastricht University, 6229 HA Maastricht, Limburg, The Netherlands; josmarmon3@alum.us.es (J.M.M.-M.); liesbeth.mercken@maastrichtuniversity.nl (L.M.); hein.devries@maastrichtuniversity.nl (H.d.V.); 2Department of Nursing, School of Nursing, Physiotherapy and Podiatry, University of Seville, 41009 Seville, Spain; joaquinlima@us.es

**Keywords:** adolescents, parents, binge drinking, risk factors, I-Change Model, focus group interviews, nursing

## Abstract

Binge drinking in adolescents is a worldwide public healthcare problem. The aim of this study was to explore the perceptions about determinants of binge drinking in Spanish adolescents from the perspective of adolescents and parents. A qualitative study using fourteen semi-structured focus groups of adolescents was conducted during the 2014/2015 school year (*n* = 94), and four with parents (*n* = 19), based on the I-Change Model for health behaviour acquisition. Students had a low level of knowledge and risk perception and limited self-efficacy. Girls reported more parental control, and when they get drunk, society perceives them worse. Adolescents suggested focus preventive actions to improve self-efficacy and self-esteem. Parents were permissive about alcohol drinking but rejected binge drinking. They offered alcohol to their children, mainly during celebrations. A permissive family environment, lack of control by parents, adolescents’ low-risk perception, low self-esteem and self-efficacy, as well as the increase of binge drinking in girls as part of the reduction of the gender gap, emerge as risk factors for binge drinking. Future health programmes aimed at reducing binge drinking should focus on enhancing motivational factors, self-esteem, and self-efficacy in adolescents; supervision and parental control; as well as pre-motivational factors by increasing knowledge and risk awareness, considering gender differences.

## 1. Introduction

Binge drinking (BD) is the most common risky drinking behaviour among adolescents, which has raised concern in many European countries [1,2,3]. BD consists of alcohol consumption that results in a blood alcohol concentration (BAC) of 0.08 g/dL or more within a period of 2 h [1,2]. In men, blood alcohol concentrations of more than 0.08 g/dL typically occur after consuming five or more standard drink units (SDUs), or standard glasses of alcohol, in about 2 h; in women, this occurs after consuming four or more SDUs in about 2 h [2,4,5]. There is a lack of consensus as to what is considered an SDU, which could be the result of the cross-country variability of criteria regarding the amount of alcohol consumption per episode [1,3,6,7], as well as a lack of consensus on the estimated average BAC in the adolescent population [5]. In this study, BD is considered the consumption of five or more and four or more SDUs of alcohol by men and women, respectively, in a short space of time, or on a single occasion [1,2,4,8,9], this definition being consistent with Spanish epidemiological data from the survey Encuesta Sobre Uso de Drogas en Estudiantes de Enseñanzas Secundarias (ESTUDES) [8]. BD has serious consequences such as brain damage, suicide, sexually transmitted diseases, violence or road traffic accidents [10].

In Europe, according to the 2015 European School Survey Project on Alcohol and Other Drugs (ESPAD), 35% of 15 to 16-year-old adolescents reported BD [11], while in Spain, the percentage stood slightly lower, at 32.2% of students between 14 and 18 years of age [8]. However, previous research has suggested a convergent trend in youth drinking cultures, since BD has increased in southern Europe and decreased in the north [12,13]. Despite these global trends of current adolescent cultures, the ways in which young people are socialised regarding drinking customs are still very different among the countries of Europe [3]. A study carried out by Rolando and Katainen [13], where they compare Italian and Finnish adolescents, established that both have very different expectations about the consequences of drinking, as well as the kinds of drinks, quantities or with whom they drink alcohol. In Spain, BD is associated with the “botellón”, consisting of drinking alcohol excessively in public places with friends during weekend nights [14,15], using it as a synonym for intoxication-oriented drinking [16].

In order to prevent BD in adolescents, it is important to know the factors associated with this behaviour. Based on the I-Change Model [17], a behaviour is the result of a person’s intentions, action plans and abilities. It also highlights the influence of motivational factors (attitude, social influence and self-efficacy) as determinants of behaviour acquisition. The person’s attitude consists of the perceived advantages and disadvantages of the behaviour. Social influences consist of the perception of others carrying out this type of behaviour (social modelling), the norms that people have with respect to these behaviours (social norms) and the perceived pressure from the environment (social pressure). Self-efficacy refers to a person’s perception of their capability to carry out the type of behaviour [18,19]. Motivational factors are determined by various awareness factors (knowledge, risk perceptions and cues to action), predisposing factors (behavioural, psychological, biological, social and cultural factors) and information factors [17].

Most of the evidence regarding the determinants of BD emerged from different methodological approaches, due to the lack of qualitative studies that delved deeper into this topic. Some qualitative studies have explored the social meanings and functions of alcohol use among young people, while also considering cultural differences [13,15,20,21]. Despite this, only two qualitative studies, one about pregnant women and their alcohol consumption in pregnancy [22], and the study conducted by Jander et al. [23] in the Netherlands, with the same target group, used an approach based on a behavioural change from the I-Change Model. For instance, previous studies, with different methodological approaches, show that BD is determined by an individual’s intention to engage in BD [23,24], their attitude and self-efficacy [24,25], as well as by social influences of people around them [22], such as best-friend norms and friend norms [26], social pressure [23,27] and friend [14,28] and parent modelling [13,23,29]. Finally, BD can also be influenced by socio-demographic factors such as age [30], gender [15,31] and socio-economic status (SES) [30,31], among others. Likewise, the Spanish context, with its own characteristics, such as the “botellón”, can lead to differences with respect to other countries [13].

Furthermore, previous research demonstrated the importance of parental influence on BD in adolescents. Currently, attitudes toward traditional alcohol-raising practices appear to be ambiguous and they are being called into question [21]. On the one hand, some qualitative research established that letting children drink at home with them can be protective against BD [20,21]. In this sense, a focus group study with parents conducted in Italy stated that allowing children to taste alcohol for the first time at home and under the parents’ supervision can be an effective way to establish a clear dialogue between parents and children, with the aim of improving shared rules and parental control [21]. Similarly, another qualitative study conducted with Danish parents established that instead of trying to ban adolescents drinking, they try to minimise the risk by helping them learn to “master intoxication” by controlling the amount and type of alcohol that they consume [20]. Instead, other studies established that high parental alcohol consumption levels [23,29,32], low parental supervision, non-rejection of BD and unstructured households [1,33], as well as less strict alcohol-related rules [34,35], were directly related to the alcohol consumption and BD of adolescents. In Spain, parental permissiveness regarding their children’s alcohol consumption is high: 48.3% of adolescents are allowed to drink alcohol by their parents [8]. As we found contradictory findings in the different studies, we highlight the importance of continuing with the research on parental views of alcohol consumption and BD in their children, which could guide the development of effective health programmes to address BD [19].

Regarding gender, Spanish girls increasingly report BD as a standard feature of their social life, and at younger ages with respect to boys [8,15], showing a diminishing gender gap in the prevalence of drinking, with equal levels between boys and girls [8,15,36,37]. A previous study showed that sensation-seeking was a noticeably stronger predictor in males, although it did not find significant differences by gender regarding coping motives [26]. Another study found that boys with high socio-economic status are significantly more likely to have BD experience, whereas girls’ BD was not found to be associated with socio-economic status [31]. Although the way of drinking between boys and girls may be different, there has been little research on the differences between girls and boys regarding the different determinants that influence both genders to engage in BD, suggesting the need for gender-specific analyses of factors associated with BD.

The objective of this paper is to explore the perceptions about determinants of BD in Spanish adolescents from the perspective of adolescents and parents, considering gender differences and whether they are engaging in BD.

## 2. Method

### 2.1. Design

A qualitative study based on Focus Groups (FG) was carried out, which took place during the 2014/2015 school year in Andalusia (South of Spain), following the COREQ guide [38].

#### 2.1.1. Sample/Participants

Fourteen FGs with students between 16–18 years of age (*n* = 94) enrolled in the 4th year of Compulsory Secondary Education (CSE) and the 1st year of Baccalaureate (equivalent to 10th and 11th grades in the USA) were created. In addition, four FGs with parents (*n* = 19) of students of the same age groups were composed.

A purposive sample was used. After contacting the head of the schools, adolescents were recruited by teachers and were told beforehand about the purpose of the study. FGs were organised based on academic year, gender and drinking behaviour. We conducted eight FG interviews with students from the 4th year of CSE, and six with students from the 1st year of Baccalaureate. Regarding gender, four FGs consisted of males, three consisted of females and seven were mixed. In three FGs, students met the criteria for BD (in the previous 30 days); in three other FGs, the students had not participated in BD; and eight were mixed.

In addition, the parents were recruited through the parents’ association. Parent FGs were composed of mothers, except for one including one father. Therefore, in general, we talk about groups of mothers. The four groups of mothers belong to four different secondary schools (Table 1).

#### 2.1.2. Recruitment Process

The researcher was introduced and explained the objective of the FG. Teachers were present only during the selection of the children.

First, the adolescents filled out a short questionnaire to assess demographic variables. Afterwards, we asked them if they had participated in BD in the past 30 days (“How often did you have 4/5 or more glasses of alcohol on a single occasion in the previous 30 days? Answer with a number”). Depending on the response, groups were assigned.

Regarding the parent group, only the researchers were present during the interviews. Initially, parents filled out a demographic questionnaire. The average duration of both adolescent and parent groups was 1 h.

Two researchers were present in all FGs. One guided the discussion until all questions were exhaustively answered. The second took notes and checked whether all question topics were covered. Both had previous experience in the subject matter and performed the subsequent analysis.

### 2.2. Data Collection

A semi-structured script, based on the I-Change Model [23], was provided. The script asked open-ended questions that allowed participants to get involved and discuss the information received and ensured the natural flow of the discussion. In addition, it allowed us to centre the discussion on what we really wanted to know. Table 2 and Table 3 show the themes, subcategories, frequency of words and concepts and sample questions for adolescent and parent FGs. Saturation criteria were considered, in order to conclude data collection.

#### 2.2.1. Data Analysis

Recordings were transcribed literally for content analysis using QSR NVivo 10 software (Melbourne, Victoria, Australia). First, all material was revised in its original form. These transcriptions were completed by comparing the interviewer and observer’s field notes. For the analysis and interpretation, the responses and conversations of each adolescent were summarised and categorised in different sections, according to the analysis categories of the I-Change Model. More than one conversation or verbatim can appear in different categories of the model, since these are not airtight compartments. In addition, adolescents’ comments were categorised according to gender, and according to whether they had participated in BD in the previous 30 days. Verbatim were classified in each of the following groups: General Verbatim (VG); Verbatim Gender Male (VGM); Verbatim Gender Female (VGF); Verbatim Binge drinking Yes (VBDY); Verbatim Binge drinking No (VBDN); and Verbatim Parents (VP).

Descriptive analyses of all participants were performed. Regarding adolescents, we researched the differences between BD and non-BD through the chi-square test for categorical variables (gender and educational level) and the Mann-Whitney test for continuous variables (age), as well as a Pearson R for the number of BD occasions and age. For the sample of the parents, only a descriptive analysis was carried out.

#### 2.2.2. Validity and Reliability/Rigour

The semi-structured script was previously used by Dutch adolescents, ensuring validity [23]. Information analysis was carried out by two members of the research team, who coded the information according to themes, in order to check the analysis process and give it a greater degree of reliability [38].

### 2.3. Ethical Considerations

The ethical considerations of the Declaration of Helsinki adopted at the 18th Assembly of the World Medical Association in Helsinki (Finland) in June 1964 were respected.

In addition, the participants were informed that the FG would be recorded on tape, only be accessible to the research team and that the processing of the answers would be confidential. Informed consent was obtained actively from parents and adolescents. The study was approved by the Bioethics Committee of Andalusia, contract PI-0031-2014.

## 3. Results

### 3.1. Adolescents’ Focus Group Interviews

#### 3.1.1. Questionnaire: Characteristics of Adolescents

Forty-nine (52.1%) participants were male, the average age was 16.46 (standard deviation (SD) = 0.81); 51 (54.3%) participants were in the fourth year of CSE and 43 (45.7%) in the first year of Baccalaureate. In addition, 38 (40.4%) adolescents had engaged in BD in the last 30 days, being 22 (57.9%) males and 16 (42.1%) females. The average number of BD episodes in the last 30 days was 1.22 times (SD = 2.75). Age (*p* = 0.008) and educational level (*p* = 0.001) were significantly associated with BD, with older adolescents associated with engaging more in BD. In addition, educational level was also associated with the number of BD episodes (*p* = 0.008).

#### 3.1.2. Binge-Drinking Patterns

The common BD age of onset was 14–15 years old, without gender differences. They related this moment to “testing”, alcohol-drinking experimentation in an environment where others were also drinking (VG):
“(alcohol drinking) My first time was at 14..”, “There were a lot of friends there in the group and they started to drink, and they seemed quite happy and I wanted to feel like them…then, they said ‘come on, nothing happens...drink a little’, and then you start to do silly things and end up drinking”.

Binge drinkers stated that they usually participate in BD with friends on weekend nights at parties, at “botellón” events, celebrations and birthdays (VBDY): *“I do not usually drink if it’s not a party...just parties...”, “at the fair, at parties, birthdays...”*, and they used to drink more than 4/5 standard glasses of spirits, i.e., rum, whisky, etc. (VBDY):
“In the past, when you drank 4 glasses you were (drunk)...and now you drink 4, 5 or 6 and you’re still not drunk”, “Maybe I’ve drunk 4 or 5… or half of a bottle”, “Rum, for example, I drink Barceló^®^ (a brand)”, “Whiskey”, “I, Beefeater^®^ (a brand of Gin)”.

However, non-binge drinkers reported to have consumed alcohol, but they did not usually drink more than one glass. There is a very similar pattern of BD between girls and boys. Although, boys reported that during the week they usually drink beers, whereas girls usually drink red wine with soda.

#### 3.1.3. Predisposing Factors

For many binge drinkers, the main way to have fun was to go out to drink with friends, instead of other healthy leisure activities. Furthermore, they said that different psychological aspects, such as shyness, low self-esteem and fear of starting relationships, directly affect BD (VBDY):
“Not for me, but for people who are very shy, they let themselves go more when they are drunk, they do other kinds of things, it takes away your fears...”, “you lose shame, that’s an advantage for shy people”, “if you do not have self-esteem, you’re going bad”, “shame is more removed and you are going to talk to people, even if you do not know them”.

Girls reported more parental control than boys and they are more socially stigmatised. Moreover, girls reported that boys engage more in BD because of their higher alcohol tolerance. Additionally, when boys had participated in BD, they tended to become aggressive and look for fights, which was less frequent among girls. Non-BD girls said that they usually feel more vulnerable to non-consensual sex or sexual assault if they get drunk (VGF):
“A drunk man looks less ugly than a drunk woman… that is what society thinks...”, “…you’re not going to rape a guy in the street, but a girl who is “too” drunk...a guy can rape her...”, “If you are drunk, 2 or 3 guys throw something into your drink and they can do whatever they want with you...”, “In addition, parents think that it is more dangerous for their daughter as she is more at risk...”.

#### 3.1.4. Awareness Factors

Binge drinkers mentioned alcohol’s disinhibiting effects, which help reduce shame; however, they did acknowledge that BD can also affect your health, for instance, damaging the liver. They also mentioned that BD consequences would probably depend on people and alcohol tolerance (VBDY): *“and that inhibits you... you lose your shame...”, “If it’s excessive... it can lead you to suffer diseases and things in your body”, “Of course it’s bad... because over time it damages your liver”*.

In the mixed group, there was no consensus on the consequences of BD. They mentioned ambivalent short-term consequences, such as fun during parties but having hangovers the next day, memory loss (from the previous day) and parental reprimands (VG): *“Then the next day it was bad”, “I felt awful”, “I felt very good, because I laughed at everything, but later when I arrived at home all the laughter was taken away...”, “…When I arrived, I was punished”*.

Both genders associated the concept of BD with daily consumption, going over the limit or losing consciousness. Yet, as none of the adolescents were familiar with the definition of BD: the concept was explained to clarify it for the remaining part of the interview. Almost all adolescents thought that that amount of glasses of alcohol was small. Some girls believed that BD depended on many factors, such as the type of drink, personal tolerance or having previously eaten (VGF): *“That consumption is normal...”, “it depends on the person’s tolerance and also the volume of alcohol you have or what you drink”, “it depends on the person and the day, whether you have eaten or have not eaten...”, “Losing consciousness, when you lose control”*.

The boys did not perceive any dangers in BD (VGM): *“…Nothing bad happened to me with the alcohol”, “If it is under control, nothing happens”*, just like the binge-drinker group. Instead, the girls referred to dangers such as an alcohol-induced coma, family damage, fights, traffic accidents, injuries, etc. as short-term consequences of BD (VGF): *“You are not aware of what happens around you, they can give you a push...and you only realise when you are already “thrown” to the ground”, “Because you can do things unconsciously...”, “The hangover”, “The headache”, “The vomit”, “Dizziness”, “can give you an alcohol-induced coma...”*, just like the non-BD groups (VBDN): *“You get more aggressive”, “fights, a lot of fights over alcohol”, “that hooks you, it can affect your life in many ways, your health, your friends”*, as well as addiction and liver diseases in the long-term (VBDN): *“That destroys you in the long term…”, “you have a good time, but then in the long run you have to pay…”, “…you become addicted…”*.

#### 3.1.5. Motivational Factors

Regarding the attitude, as advantages, binge drinkers stated that BD could help to socialise, flirt or dance (VBDY): *“I do not dance like this, I’m incapable... I drink until I’m tipsy and then I dance”, “well, to socialise, if I use it...”*. They also highlighted emotional factors such as having fun, laughing, feeling adrenaline, etc. (VBDY): *“It gives you an adrenaline rush…”, “having fun with friends… being silly, and we laugh a lot”*.

As disadvantages, they highlighted fights, aggressiveness, memory loss, losing keys or wallets, falling to the floor, or doing things that they later regret (VBDY): *“Things are done without thinking”, “and then you regret it, because you don’t know what you’re doing. Imagine you’re in that state and you decide to drive the car”*.

However, non-binge drinkers saw no benefits to BD (VBDN): *“It really does not help you at all”, “The truth is that there are no advantages”, “They are inconveniences more than anything…”*. As disadvantages, they mentioned dizziness, vomiting, disorientation, memory loss, addiction or even death. They also mentioned that sometimes they are involved in problems, such as fighting when their friends get drunk. In addition, they pointed out that being drunk does not help them flirt (VBDN): *“It brings very bad diseases, I have an uncle who has a liver disease and is dying “, “The hangover “, “The headache…”, “The vomit “, “Because if they get into a fight and you are not drunk, they can involve you in the fight, they get you into trouble”*. No differences in these beliefs were found regarding gender.

Concerning social modelling, some non-BD mentioned that their families (parents and siblings) do not usually drink alcohol, although some friends have engaged in BD, whereas binge drinkers indicated that their friends and family (parents, siblings, uncles/aunts and cousins) often drank alcohol (VBDY): *“I drink with my sister”, “oh well, and with my cousin too”, “My parents and friend”, “My sister, but she drinks like me”*. There is no difference regarding gender.

In relation to social norms, in general, students emphasised that people who would not approve of their drinking were those who love them and older people, for instance, grandparents (VG): *“all the people who love you, they will not want you to drink”, “my grandparents would tell me not to drink”*. Moreover, they said their best friends think they should not drink, while other friends think that they should, differentiating between their most intimate and other friends (VG): *“It depends on the type of friends you have. If it’s a friend who really loves you, when you’re bad with alcohol they will tell you to stop”, “my closest friends think I should not drink”*.

In addition, students thought that there was a certain alcohol-related permissiveness in society (VG): *“It is something that is widely accepted nowadays”, “… it is very “legalised” today”*. Many students from the mixed group said that parents, siblings, cousins and friends think that they should drink, but they do not approve of BD (VG): *“he always tells me... do not drink too much...”, “they let me drink as long as I do not return to my house completely out of my head”*. No difference was found regarding BD and gender.

Binge drinkers did not perceive any pressure from their peers regardless of the situation. However, non-BD perceived pressure from peers, mostly when they were with a large group, at a party, “botellón”, etc. (VBDN): *“If pressured, I would say yes, for friends”, “When I am at a ‘botellón’ and I do not feel like drinking, they say “well, then why do you come to the ‘botellón’?” and then I drink”, “…they say let’s go, and call me boring or dull”*.

Furthermore, the group also affected the amount of alcohol consumed, the type of drinks and the higher probability of BD (VBDN): *“I think that what people do is also very influential. If everyone drinks, I have to drink, because I will not be left behind”*.

The pressure to drink is greater in younger students, being a significant trigger for the onset of drinking (VBDN): *“The youngest ones feel more forced to start to drink. They see the older ones, and they want to grow up fast”, “the first time it is normal to be told to drink”*.

Binge drinkers mentioned having been offered alcohol by a family member or their parents when they were at a family celebration, such as a wedding, fair, birthday or in a bar, etc. (VBDY): *“Yes, when my mother takes us from time to time to a bar and she says, do you want some red wine? And she offers it to me”, “maybe beer with my father too, if he has a beer, he gets one for me”*. No difference was found regarding gender.

Concerning self-efficacy, all adolescents found it more difficult to say “no” when they are at drinking places, events such as a “botellón”, celebrations, parties or at a disco (VG): *“When you go to the disco, or to the ‘botellón’…you know that what you do there is drink”, “at a ‘botellón’ it is more difficult...”*. Some binge drinkers said they had never considered it difficult to say “no” to drinking. However, non-BD found more difficulty with a larger group of friends, and it was easier when they were at home or at a friend’s home (VBDN): *“I think it is more difficult when you are with more people. When you are alone you say: “Why, if I do not want to” but with people you say: “Go on then”“, “Why am I going to drink at home alone... to look in a mirror and imagine people are there?”, “it is easier not to drink at a friend’s house…”*. No difference was found regarding gender.

#### 3.1.6. Family Factors

Most students outlined not experiencing much strictness from parents regarding drinking, and mothers are even less strict (VBDN): *“My mother perhaps does not punish me too much for arriving an hour late”*, although parents know at all times with whom their children go out, and even know that they and their friends drink alcohol. Non-BD claimed their parents to be stricter with them (VBDN): *“my father is stricter”, “They like to know where I am and who I am with…to control what I’m doing and what I’m not doing”*. Binge drinkers revealed that they had permissive parents. They can easily disobey the rules, such as being home late and/or drunk with scarce reprimands or punishments (VBDY): *“They tell me today half an hour less, but then, I arrive at the same time”, “they scold me, they tell me “next time”, but then nothing happens”*. In contrast, girls and non-BD reported that if they arrived home drunk, their parents would get angry, punish them, lose faith in them and not allow them to go out again (VGF): *“my mother is like that...she gives me a slap that leaves me like...”* (VBDN): *“Man...my mother sees me coming through the doors and if I fall on the floor...my mother does not let me out anymore or give me more money...”*. Yet, these consequences would be unlikely for boys when arriving home drunk.

Binge drinkers reported that their parents know that they drink alcohol and allow it, although they do not approve of BD (VBDY): *“My parents, when I am going to a party…they say...do not drink too much”, “They know I drink”*.

In general, students said that their parents usually drink alcohol at home but not excessively (a glass of wine or a beer, usually accompanied by food), their father being the one who drinks more beverages with a higher alcohol content (VG): *“My father drinks beer on weekends, when we go out to eat”, “my mother drinks beer without alcohol and my father regular beer”, “they drink beer...my father ‘cubata’ (spirits), but my mother does not”*.

The boys reported on the permissiveness of their parents. They stated that they have consumed alcohol in front of their parents and that parents allowed this. Parents tell them not to drink too much. The boys also revealed that, during celebrations, parents and family members offered them alcohol, and parents allowed them to drink. This was also reported in the conversations of binge-drinker groups (VGM): *“Maybe they’ll have a beer and they’ll grab one for me”, “in a bar, they ask me: what do you want?”* (VBDY): *“My father gives me a beer”*. However, the girls reported that they had never been offered alcohol by their family (VGF): *“My mother has never offered me (alcohol) but has never told me not to drink either”*.

In general, parents do tell them about the consequences of alcohol, but they allow them to make their own decisions (VBDY): *“My father has never forbidden me to drink, he has taught me that it will not bring me anything good, you know what you have to do and what not”*. Adolescents reported that parents have spoken to them about alcohol and its consequences. Some of the non-BD boys said that their fathers alluded to doing sports to prevent drinking, smoking or using drugs (VG): *“My father has always told me about alcohol, what can happen”* (VBDN): *“I love sports, because my father always tells me that athletes do not drink, or smoke, or anything, they can be role models, that helps me a lot”*.

#### 3.1.7. Ability Factors

Some binge drinkers said they only drink if they want to, so they proposed some alternatives to avoid BD. Instead, non-BD proposed doing alcohol-free activities and avoiding situations where they could be pressured into drinking (VBDN):
“if it’s a ‘botellón’ I do not go...but if it’s a party or a birthday I go...why do I have to drink?”, “I am by the river with my friends and we spend the day playing football, cards, etc.”, “Being with your friends, but not going to a party, for example, being at home with your friends and ordering a pizza, watching a movie...”, “bowling…”, “or say that you do not drink”.

Some said they do not usually drink because they practice sports and it would affect their performance, and their teammates did not usually drink. They also proposed elaborate strategies, such as improving self-esteem and self-confidence, losing one’s shyness to be able to relate to people without drinking and counting on the support of non-drinking friends and family (VBDN):
“An alternative is to be sure of yourself, that you do not need alcohol to do what you want to do”, “Loving yourself”, “be in an environment of trust... feel good in the place where you are... you do not need anything more”, “look for people who think the same as you…”, “be sure of yourself and of what you want”.

Furthermore, some girls said they drink less when they go out with their boyfriends, as well as when they have an exam the following day.

#### 3.1.8. Intention Stage

Most adolescents had no intention of changing their behaviour. We found a negative precontemplation among binge drinkers (VBDY): *“Saturday we will go out to drink”, “this weekend we have a party”, “I have a barbecue tomorrow and some beer will be had”*, whereas we found a positive precontemplation among non-BD (VBDN): *“I have a party to attend but I do not think I’ll drink…”, “we have tests, I do not usually go to parties to drink”, “we prefer drinking juice or eating sweets and playing cards in the park, more than drinking alcohol…”*. No difference was found regarding gender.

### 3.2. Mothers Focus Group Interviews

#### 3.2.1. Characteristics of Parents

Eighteen parents (94.7%) were female and the average age was 46.5 (SD = 2.5). Regarding educational level, nine (47.4%) had university degrees, five (26.3%) had vocational training and five (26.3%) were secondary school graduates.

#### 3.2.2. Awareness Factors

Mothers did not know the definition of BD, believing that it is prolonged and chronic consumption that causes long-term health problems (addiction and developmental impairment). In addition, some of them mentioned short-term consequences, such as disinhibition, needing emergency care or entering an alcohol-induced coma (VP): *“…it causes addiction”, “delays certain levels of physical and mental development”, “there are many who drink excessively at a ‘botellón’, and in an ethylic coma you can be done for (die)”*. Some of them thought that girls are more vulnerable than boys (VP): *“my daughter told me that someone put something in a girl’s drink, and the girl was totally uninhibited and started to flirt with the boys”*.

Moreover, some mothers considered the social influence of the consumer society and the major economic interests associated with BD, which pressure adolescents to drink alcohol (VP):
“The consumption of alcohol is very normalised”, “consumer society pushes young people”, “because that brings a lot of money and it’s a market of interest…”.

In fact, in some towns there are places designated for drinking. Other mothers highlight the influences of the group of friends on drinking/BD (VP):
“The group pressures you and it depends on your level of maturity to say yes or no”, “the opinion of a friend weighs more than that of a father”.

Finally, some mothers believed that their children had not started drinking or never would, so they are not at risk, although many classmates and even younger adolescents had started (VP):
“my kids, as far as I know, have not yet tried alcohol, but the kids around them… it’s crazy, I have seen children with ‘litronas’ (big bottles of beer) and tobacco, the same age as my kids…”.

They consider it normal that their children do not recognise the consequences of alcohol consumption, due to the nature of adolescence (VP):
“I have been a teenager, and I was not aware of dangers... I am aware now that I have children”.

#### 3.2.3. Attitude Towards Adolescents’ Drinking and Binge Drinking

Most mothers do not allow their children to binge-drink; they prefer their children to have a healthy life (VP): *“them being athletes is the best”*. They thought that drinking is not necessary to have fun (VP): *“…you do not have to drink to have a good time”*. Others did not consider it bad that their children drink alcohol moderately, considering that giving access to alcohol at home could prevent them from going to the “botellón” (VP):
“But normally he gets drunk when he goes out”, “You can do it with control”, “excess is what makes it bad”, “They met in my house and they had a little party. Their father and I went out and we left them at home, there were 6 or 7 of them and the rest of the class went to the ‘botellón’”.

#### 3.2.4. Rules and Norms

More than half of the mothers did not have clear rules, and said they cannot control their children’s drinking, departure or arrival times, the group of friends with whom they spend time, etc. (VP):
“I give him the freedom that my mother gave me, with moderation”, “She is the one that establishes the curfew”, “there are times I exercise more control and times when I am less strict”, “in high school you lose a little (control), because you do not know the kids or their parents as you did at school”.

Others said they are stricter regarding their children’s schedules, and try to know with whom they spend time (VP): *“I do not think that my 15-year-old daughter has to go out until 12”, “when she goes out I want to know where she is”*. In addition, one mother said that they did not treat sons and daughters equally, showing gender bias, being more protective and stricter with the daughters, even having the son take the sisters home (VP): *“My son comes home later because he is a boy...and he picks up his sister”*. Most mothers showed permissiveness, allowing their children to drink at home, and even offering them alcohol (VP): *“my son can drink at home if he wants...and out”, “I’ve told him ‘of course’, and he’s taken a sip and tasted it”*. Some mothers thought that forbidding their children from going to the “botellón” or from drinking alcohol could be worse because it could encourage them to do so (VP): *“I think that everything that is prohibited is what makes you want to do it”*.

They reported few reprimands when their children disobeyed a parent’s rules, such as if they came home drunk (VP): *“I would not scold him either, because in those situations there is no point in scolding them”*.

#### 3.2.5. Beliefs about Parental Influence

Some mothers thought that it is a family’s responsibility to educate and inform their children about the dangers of drinking and BD (VP): *“They have to be informed, with the basic work done in the family, that they have other activities and other alternatives to the ‘botellón’”*. However, others stated that it was not only family who influences adolescent behaviour, but also the consumer society, as was previously indicated.

#### 3.2.6. Action Planning: Abilities and Alternatives

More than half of the mothers highlighted that adolescents had no alternatives, since society and groups of friends imposed the “botellón” as a socialisation mechanism. Moreover, some mothers stated that drinking does not interfere with adolescents’ studies, leisure or life, provided that they drink with “control” (VP): *“There does not have to be an alternative, they can do it with control”*. However, many mothers would prefer that adolescents substitute the “botellón” with other leisure activities.

There should be greater awareness about alcohol, with more emphasis on training adolescents to say “no” (VP): *“…having other enriching activities, other alternatives to the ‘botellón’ when they go out with friends”, “Learning to say “NO”, that must be taught…”*. They said that talks about alcohol at school were ineffective, so they proposed to present real cases in order to shock adolescents (VP): *“Even if there are talks, there should be other things”, “I would show them more in real life…”*.

#### 3.2.7. Parent-Child Communication

Almost all mothers stated that they have good communication with their children, and that they talked about alcohol. They thought that if their children had any problems, they trusted them, although they stated that their children are reluctant to talk about certain topics (VP): *“We trust each other”, “I talk to her about the ‘botellón’…”*. Others claimed their children may not tell them the truth about their drinking behaviour: *“What happens is that you can talk to them, but you never know if they’re telling you the truth or not”*.

## 4. Discussion

The main purpose was to explore the perceptions about determinants of BD in Spanish adolescents from the perspective of adolescents and parents, taking into account gender differences and whether they are engaging in BD. To our knowledge, this is the first in-depth study on the determinants of BD in Spanish adolescents from the perspective of adolescents and parents, based on the I-Change Model.

Regarding BD patterns, this study is in line with other works carried out in Spain and Europe, where BD shows positive representation among adolescents [1,39]. In addition, some factors, such as intensity and frequency of BD, seem to determine its consequences [39]. Currently, the ways in which young people are socialised in terms of consumer habits are very different among European countries, although there may be certain global trends and cultures [3]. Culture plays the main role in shaping the way people learn to drink and establish their alcohol consumption patterns, as well as their problematic alcohol-related behaviours [21]. Despite this, BD is no longer associated with certain cultural events, being a regular activity for many adolescents [27]. In our study, many students are often involved in BD at public places, mostly on weekends, drinking mostly spirits with their friends (“botellón”). Therefore, accessibility to alcohol, and planning of spaces and leisure time, can affect BD [14,15].

A previous study with parents claimed that adults’ opinions regarding to the alcohol consumption of adolescents seem to some extent to be affected by an excessive alarm spread by media messages [21]. In our study, mothers claimed that BD is institutionalised, with economic interests that pressure adolescents to drink alcohol [8]. In Spain, although only legally purchased and consumed from the age of 18, alcohol is provided to adolescents, and the “botellón” phenomenon shows the limitations of the law [40]. Consequently, not only legislation, but socio-political and environmental factors should be addressed, as well as the development of alternatives for leisure activities and free time for adolescents [14,15].

Gender can be considered a predisposing factor for BD; girls have more gender stereotypes and a stigmatised negative social image regarding drinking, with a higher risk of being exposed to non-consensual sex. In this line, in a previous study with focus groups, authors claimed that both girls and boys are reluctant to associate women with a variety of alcohol-related activities, including being intoxicated. Besides, this sexual stereotyping appears present at a relatively young age, and continues through adolescent experimentation with alcohol [20]. In our study, girls reported more parental control than boys, as well as more punishments if they got drunk [14,15]. Different studies show that boys tend to drink alcohol more heavily [1,40]. However, recent evidence suggests a diminishing gender gap, with equal levels of consumption between both genders [8,14,15], as indicated in our study. Furthermore, an increase in female alcohol drinking, together with the vulnerability to which they are exposed, requires greater efforts to address drinking problems according to gender [41].

Adolescents highlighted low self-esteem as an important factor in BD, using alcohol to socialise, to try to “fit in”, as well as to lose fear and shame. In this sense, some authors suggest that the needs to belong to a group and develop self-esteem may both be related to student drinking [42]. Much of human behaviour is motivated by a desire to satisfy needs of belonging, as feeling accepted and fitting in with those around them is a fundamental need [42,43]. Furthermore, self-esteem may influence how individuals satisfy their connectedness goals within specific relationships [44]. Self-esteem refers to the degree to which individuals value, respect and accept themselves, and is widely considered to be one of the critical elements that affect future behavioural choice and action [45,46]. Hamilton et al. [42] concluded that students with low self-esteem are drinking because they lack the self-resources to deal with unmet needs of belonging. These findings suggest that low implicit self-esteem may be a risk factor for student drinking.

Similarly, Bartsch et al. [41] found that youths with low self-esteem have more risk with respect alcohol consumption and BD, confirming the protective role of high levels of self-esteem against substance use for adolescents [46]. Despite this, other studies did not find links between reported self-esteem and substance abuse [47,48], which may be due to different target populations, settings, covariates and measures of substance use.

Regarding the cognitive factors, we found there is little knowledge about BD in both adolescents and mothers, as in the study by Jander et al. [23], binge drinkers indicated that the amount of alcohol that was defined as BD was low, depending on the “tolerance” of each person. The need to provide health communication messages to prevent misconceptions is thus clearly demonstrated. Moreover, although many adolescents knew that prolonged consumption could cause long-term health problems, the short-term consequences, such as losing control and falling, hangovers, etc., were considered superfluous. Previous studies indicated that this could be due to low risk perception in adolescence [33]. In this sense, we also found a difference between both genders, since boys mention more advantages of BD with a lower risk perception than girls [49]. Instead, the girls referred to dangers, such as an alcohol-induced coma, family damage, fights, traffic accidents, injuries, etc., as short-term consequences of BD.

Similarly, some mothers did not perceive any risk of alcohol consumption in their children, as they believed that they had not started drinking or binge drinking. In summary, our data reveal quite alarming low-risk perceptions concerning alcohol in both adolescents and parents [32,50]. In contrast, Rolando et al., [21] claimed that adolescents were active actors whose drinking, like adults’, is based on a conscious and rational choice, and they even believe that adolescents are more aware of this risk than adults.

According to the National Drug Plan [8], BD is tolerated by Spanish society, and even approved of. In this sense, adolescent BD could be perceived as an inevitable and standardised process. In addition, although non-binge drinkers did not find advantages in engaging in BD, binge drinkers did. Hence, from a public health perspective, we should be concerned about the positive cultural aspects that adolescents associated with BD, and consider encouraging awareness and stigmatising alcohol drinking/BD. This fact highlights the importance of changing attitudes, especially in those who have a greater disposition towards drinking, emphasising the disadvantages.

Regarding social influences, similarly to previous studies, alcohol drinking and BD are influenced by both family behaviour [32,51] and peer-group behaviour [3,23,51]. Furthermore, it seems that there is a relationship between the size of the peer group and BD [1,23]. The pressure of the group can affect not only the amount of alcohol, but the type of alcoholic drinks consumed as well [27]. In contrast, sometimes peer influence can prevent youths from drinking/BD, such as the case of the best friends of non-binge drinkers, who may tend to worry more about the drinker’s health [52].

Furthermore, students identified “botellón” events or parties such as fairs, Holy Week or Christmas as the most difficult times to avoid drinking, showing low self-efficacy in these situations [53]. Thus, health communication messages should focus on strategies and alternatives to cope with these difficult situations, as well as on reducing the influence of friends or family members, strengthening self-efficacy and providing them with action plans for managing risk situations satisfactorily.

Parental consumption, absence of parental control and communication, were found as family factors associated with BD in adolescents [29]. In our study, alcohol drinking often takes place in a family context, being a more typical behaviour of families in southern Europe than their northern coevals [13]. Adolescents and mothers reported the availability of alcohol at home, and many students answered that their family had offered them alcohol. The mothers’ attitude was one of rejection of BD, while there was greater permissiveness with respect to moderate alcohol consumption. In this sense, in Spain, 48.3% of parents allowed minors to drink alcohol [8]. Furthermore, as in the study of Jander et al. [23], some mothers do not have clear rules and say they cannot control their children regarding drinking. Moreover, many mothers lack the self-efficacy required to control their children; they even think that forbidding them from going to the “botellón” or not drinking is harmful, because it is a way of encouraging them to do the opposite. In addition, we found that binge drinkers stated that their parents were not too strict, whereas non-binge drinkers claimed that their parents were stricter with them. Furthermore, some differences between genders were found, since boys stated that their parents allowed them to consume alcohol with them, and that they even offered them alcohol. However, the girls reported that they had never been offered alcohol by their family. In this sense, girls reported more parental control than boys. Previous studies have found association between high levels of parental control and lower alcohol drinking in adolescents [35,54]. However, Rolando et al. [21], state that the traditional alcohol-raising practices, based on applying strict rules, are currently being called into question. In their findings, it was shown that parents are more and more convinced that it is better to socialise their children to alcohol at home [20,21], which could help children to learn to “master intoxication” by controlling the amount and type of alcohol that they consume [20]. In addition, despite the low level of knowledge observed about BD, both adolescents and parents stated that they talk about alcohol and its consequences at home [55]. Despite the importance of family communication, a precondition should be to increase parents’ awareness about the problem of alcohol drinking and BD in their children, as well as of their influence on this behaviour in order to offer a reliable model.

Regarding abilities and action plans, participants identified different alternatives and strategies to avoid BD in various stressful situations. Non-binge drinkers proposed increased self-confidence, self-esteem, and support from friends to avoid BD. Quality social interactions with peers are a key contributor to positive self-esteem, a pivotal building block to healthy development [56]. Hence, health communication messages that address self-esteem, the quality of relationships and social support to avoid BD are recommended [41]. Moreover, social support could improve perceived self-efficacy, which is the adolescent’s ability to resist consumption, and to initiate behavioural change [48,53].

Finally, regarding the intention stage, most adolescents were in the precontemplation phase. According to the I-Change Model, pre-motivational factors (awareness factors such as knowledge, cues to action and risk perception) need to be addressed first, to make adolescents sufficiently aware of the seriousness of the problem [18]. Thus, future health programmes aimed at reducing BD should include pre-motivational factors.

## 5. Limitations and Strengths

Limitations: (1) Group pressure may restrict some participants and limit confidentiality. Therefore, it was explained that the data was going to be treated confidentially, and the researchers intervened only to launch the discussion. Moreover, the adolescents were recruited in the same class, so them knowing each other could condition their speech. One threat in group discussions is that some members may dominate discussions. Yet, as the researcher who chaired the discussion invited all members to freely mention all their opinions, we believe that we minimised this threat. Furthermore, as slightly more girls than boys participated in discussions, a slight overrepresentation of themes mentioned by girls could have occurred. However, the sample was still sufficiently diverse, and the setting allowed males to express their opinions as well. (2) Given that the participants and the researchers were not blinded, this fact could influence the way in which the data is presented. (3) Due to the characteristics of the sample, it was difficult to form a homogeneous group. However, it was possible to form groups according to gender and whether they had engaged in BD previously, as well as according to their academic years. The number of participating parents in groups was slightly lower than recommended, which could potentially reduce the level of interactivity. Despite this, the interactions between parents were very lively. As with other research among parents [23,57,58], most participants were women, and thus it reflects the mothers’ points of view. (4) Some FGs were performed after holiday periods (Holy Week), which would increase the number of adolescents who reported BD, but we tried to increase the number of groups in other periods, to cushion this effect.

One of the strengths of this study is that both parents and adolescents’ perceptions on alcohol consumption and BD were collected, making it possible to know the views of both groups about the determinants. Additionally, our study used a comprehensive theoretical model, as the Spanish context may lead to differences in (the importance of) certain determinants. For instance, this was also shown by the importance of fairs or the permissiveness of their parents.

## 6. Conclusions

This study reports the perceptions of adolescents and mothers about determinants of BD in Spanish adolescents, based on the I-Change Model, and how families manage their children’s drinking behaviour. The findings can help Nurse Practitioners and healthcare professionals who work in Primary Care and School Health, by developing health communication programmes to reduce BD. Our findings highlight that both adolescents and parents reported on gender differences, since the awareness factors, the social influence of the family and the people around them, attitude towards consumption and self-efficacy between others could vary according to gender. For example, boys list more advantages of BD, did not perceive any danger in BD, recognise fewer short-term consequences of BD, and reported less parental control than girls, among other features. In addition, the role of self-esteem and self-efficacy was highlighted, as was the importance of increasing the risk perception. This could be achieved, for example, by showing situations where adolescents feel identified, which could attract their attention, improve their awareness and favour the pre-motivational and motivational components of adolescents to decrease BD. Furthermore, our findings highlight the importance of increasing parents’ awareness factors, communication skills, adequate family influence, parental supervision and control of adolescents, and thus reduce BD. In summary, we believe that these results could help to design health communication advice.

## Figures and Tables

**Table 1 ijerph-17-03551-t001:** Characteristics of adolescent participants in the focus group interviews.

Sample Characteristics				Binge Drinking		
				Yes *n* (%)	No *n* (%)	*p*
Adolescents	**Variables**	FG ***: 14	(*n* = 94)			
	Age (mean, SD)		16.46 (0.81)	16.74 (0.76)	16.27 (0.80)	0.008
	Gender					0.357
	Male	4	49 (52.1%)	22 (57.9%)	27 (48.2%)	
	Female	3	45 (47.9%)	16 (42.1%)	29 (51.8%)	
	Mixed group	7				
	Education Level					0.001
	4th CSE *	8	51 (54.3%)	13 (34.2%)	38 (67.9%)	
	Baccalaureate **	6	43 (45.7%)	25 (65.8%)	18 (32.1%)	
	Binge drinking					
	Yes	3	38 (10.4%)			
	No	3	56 (59.6%)			
	Mixed group	8				
Parents	**Variables**	FG ***: 4	(*n* = 19)			
	Gender					
	Male	0	1 (5.26%)			
	Female	3	18 (94.74%)			
	Mixed group	1				
	Educational Level					
	Elementary		5 (26.32%)			
	Vocational Training		5 (26.32%)			
	University		9 (47.36%)			

* 4th CSE = acronym Compulsory Secondary Education (equivalent to 10th grade in the United States of America). ** 1st Baccalaureate: (equivalent to 11th grade in the United States of America). *** FG: number of focus group. SD, standard deviation.

**Table 2 ijerph-17-03551-t002:** Themes, subcategories, frequency of words and concepts, and sample questions for the adolescent FG interview.

Theme	Sample Questions	Subcategories	Frequency of Words and Concepts (*n*)
Binge drinking patterns/first binge drinking experience	Do you drink alcohol? How many glasses? In which situations do you drink/binge drink? With whom do you usually drink/binge drink? When was the first time you drank/binge drank and why? What consequences did you experience?	• First drinking/binge drinking episode.	Testing/experimentation (8); get drunk (10); Psychosomatic alterations (18); disinhibition (5).
		• Age of onset.	14–15 years old (20).
		• Occasions, places and days when drinking/binge drinking.	Weekend nights (12); “botellón” (12); celebration (40); birthdays (11); fairs (17).
		• Type of drinks, quantity.	4–5 glasses (15); bottles (15); rum (9); vodka (12); whisky (11); beer (13); spirits (30); wine with soda (8).
		• People with whom you usually drink.	Friends (13); Siblings (6); Parents (3); Consumer environment (4).
Predisposing factors	Do you think that the drinking/binge drinking behaviour in boys and girls is similar or different?	• Gender: stigmatisation of girl.	Parental control (4); socially stigmatised (9); more vulnerable (4); sexual assault (5); aggressive boys (5); boys’ tolerance (3).
Awareness	What do you know about binge drinking? What kind of dangers do you perceive of binge drinking?	• Knowledge about binge drinking. • Risk perception.	It is bad (23); long-term consequences (7); short-term consequences (27); disinhibition (6); damaging the liver (3); BD depends on several factors (18); BD is normal consumption (17); reduce shame (3); tolerance (9); memory loss (8).
Motivational factors: • Attitude • Social influence • Self-efficacy	What are the advantages or disadvantages of binge drinking? Which people around you usually drink/binge drink (friends, boyfriend/girlfriend, siblings, family, etc.)? Which people approve/disapprove of you drinking/binge drinking? Have you ever felt pressured to drink/binge drink? In what situations do you think it is difficult to say NO to binge drinking (at a party, at a bar, at a friend’s house, in a public place, etc.)? How difficult is saying NO in such situations?	• Attitudes: Pros.	Reduce shame (28); disinhibition (9); flirt (6); having fun/laughing (28); socialise (6).
		• Attitudes: Cons.	Disinhibition (7); lose consciousness (7); damaging the liver (8); hangovers (5); memory loss (10); vomiting/headache (5); addiction (2); ethylic coma (7); traffic accident (3); affects social life (5).
		• Social modelling.	Drank alcohol: friends (30); parents (15); siblings (5); uncles (5); cousins (4); beer (17).
		• Social norms.	Disapprove: parents (15); grandparents (13); best friend (9). Approve: friends (15); cousins (4); siblings (10); permissiveness in society (2).
		• Social pressure.	Pressure of friends (12); Group size pressure (12); no pressure (20); “botellón” (5).
		• Self-efficacy.	Ability to say no (16); difficulty saying no (9); never considered (3); “botellón” (25); celebration (25); group (7); at home (3).
Family factors	Do your parents know where and with whom you are when you go out? What do your parents think about you drinking/binge drink? Are you allowed to drink alcohol at home? Do your parents drink alcohol at home? Have your parents ever offered alcohol to you? In which situation? Have you ever talked with your parents about drinking or how much you can drink?	• Parental influence in alcohol drinking/binge drinking: Supervision and control.	Reprimands (7); supervision (6); permissiveness (18); control (7); strict (7); punishments (3).
		• Parental influence in alcohol drinking/binge drinking: Behaviour and rules.	Beer (8); wine (9); parental consumption (15); BD is not approved (4); permissiveness (19); alcohol offered by the family (35).
		• Parental influence in alcohol drinking/binge drinking: Communication.	Alcohol (12); tobacco (3); other drugs (8); consequences (5).
Ability	What do you do when you do not want to binge drink? How do you handle the situation? Which alternatives are there to prevent binge drinking?	• Action plans. • Abilities. • Alternatives. • Low self-esteem.	Resistance capacity (14); low resistance (3); leisure activities (14); self-esteem (6); self-confidence (3); social support (3); self-security (2); sport (17).
Intention	Do you have any plans for this weekend (party, birthday, “botellón”, etc.)? If so, are you going to drink/binge drink? Are you planning to quit or reduce drinking/binge drinking?	• Precontemplation. • Contemplation. • Preparation.	Drink in the future (17); not drink in the future (15).

**Table 3 ijerph-17-03551-t003:** Themes, subcategories, frequency of words and concepts, and sample questions for the parent FG interview.

Theme	Sample Questions	Subcategories	Frequency of Words and Concepts (*n*)
Awareness	What do you know about drinking/binge drinking and its short- and long-term consequences? What dangers do you perceive of binge drinking in adolescents?	• Knowledge about alcohol drinking and binge drinking.	It is bad (8); disinhibition (5); addiction (2); permissiveness in society (8); short-term consequences (4).
		• Risk perception.	Low risk perception of adolescents (3); economic interests (3); group of friends (7); think their children do not drink (17).
Motivation	What is your opinion about drinking/binge drinking in adolescents? Which are the advantages or disadvantages?	• Attitudes: Pros.	Socialise (4).
		• Attitudes: Cons.	Disinhibition (3); addiction (2); chronic problems (2); ethylic coma (7).
	What are the rules at home with respect to your children’s arrivals and departures (arrival time, with whom they go out, what they are doing)? What rules do you have at home about drinking? Are your children allowed to drink at home? What do you do when your child drinks/binge drinks? Are there any consequences for him/her?	• Rules and norms. • Parental supervision and control. • Consequences of alcohol consumption of their children. • Parental behaviour.	Permissiveness (33); offers alcohol (3); no clear rules (5); supervision (6); control (4); strict (6); parental control of girls (3); no prohibition (6); punishments (3); few reprimands (9).
	In what way do you think that parent’ alcohol consumption influences on kids?	• Beliefs about parental influence.	Permissiveness (3); normalised (10); does not allow consumption (3); permissiveness in society (23); drink at home (7); beer (6); wine (8); family responsibility (10); does not influence (5).
Action planning	Which actions could help to prevent drinking/binge drinking in adolescents? What are the alternatives to drinking/binge drinking? How could the prevention of drinking/binge drinking in adolescents be improved?	• Action plans. • Alternatives to consumption. • Beliefs about prevention of drinking/binge drinking.	No alternatives (8); training to say no (7); prevention (10); prohibit the sale of alcohol to minors (3); leisure activities (4); house party as an alternative (4).
Communication	Have you ever talked with your children about drinking/binge drinking and its consequences?	• Communication between parents and children.	Communication (9); trust (6); no communication (5); tell the truth (9)

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
