# Peer review of "Why are Spanish Adolescents Binge Drinkers? Focus Group with Adolescents and Parents"

_ijerph, 2020, doi:10.3390/ijerph17103551_

Round 1

Reviewer 1 Report

The author has done an excellent/comprehensive attempt to address the concerns of the original reviewers.   I was particularly impressed by the way the changes were particularly clear for the reviews.  I have no further comments to make.

Author Response

Comments and Suggestions for Authors

The author has done an excellent/comprehensive attempt to address the concerns of the original reviewers.   I was particularly impressed by the way the changes were particularly clear for the reviews.  I have no further comments to make.

We appreciate very much the positive comments of the reviewer 1.

Reviewer 2 Report

I found the article improved and overall I think it has reached a form that is almost ready for publishing.

I have only two main conrcens that I leave to the authors as further observations that can be evaluated by the editor.

The first is about the previous literature. 

p. 2 lines 74-78. I continue to see a lack of acknowledgement about qualitative studies that previously address this topic, and particularly parents' view on youth drinking. If it is true that only a few studies are based on a behavioural change from the I-Change Model, nonetheless the factors under study that pertain to the model were actually studied in several studies. A better knowledge of this literature could be helpful for more interesting (and updated) discussion and conclusion.

Example of literature based on parents' opinion on youth drinking and collected with qualitative methods are listed.

Østergaard, J. (2009). Learning to become an alcohol user: Adolescents taking risks and parents living with uncertainty. Addiction Research and Theory, 17, 30–53.

Rolando, S., Beccaria, F., Petrilli, E., & Prina, F. (2014). Adults’ views of young people's drinking in Italy: An explorative qualitative research. Drugs: education, prevention and policy21(5), 388-397.

(BTW the latter study was suggested in the previous review, but was apparently misunderstood and confused with another publication (Rolando and Katainen) which was actually on alcoholism and it is not correctely quoted on P. 8 line 501 (it is not that italian and finnish young people do know much about BD...) 

2) Conclusions

Conclusions are not very specific or new (it well known that prevention should be gandered or that it should aim to improve self-exteem). In my opinion, conclusions and dicussion should better highlight what the approach based on the I-Change Model add to the previous knowledge. 

p. 10 lines 602-604:"We should also focus on promoting self-esteem and self-efficacy as well as showing real cases in order to shock adolescents into awareness, and favour pre-motivational and motivational components of BD in adolescents" --> I would recommend to avoid to give such a suggestion, since it is well settled that the "fear-appeal approch" does not work in prevention!!

If you want to give practical advice to prevention, you should also deal with prevention litarature. Otherwise, keep the discussion and conclusion more focused on your theretical approach and what it adds to knowledge about the topic under studies and how it can be used to improve interventions.

General observation: here and there results are defined as objective, eg in the conclusion: "This study reports the determinants of BD in Spanish adolescents". This can sound misleading and you you should pay attention to clarify that you have investigated perceptions and opinions about determinants, rather than determinants.

Author Response

Comments and Suggestions for Authors

I found the article improved and overall I think it has reached a form that is almost ready for publishing.

I have only two main concerns that I leave to the authors as further observations that can be evaluated by the editor.

We appreciate the positive words of reviewer 2 about the improvement of our article as well as the constructive comments. We are convinced that the reviewers’ comments have improved the quality of the paper.

The first is about the previous literature

P.2 lines 74-78. I continue to see a lack of acknowledgement about qualitative studies that previously address this topic, and particularly parents' view on youth drinking. If it is true that only a few studies are based on a behavioural change from the I-Change Model, nonetheless the factors under study that pertain to the model were actually studied in several studies. A better knowledge of this literature could be helpful for more interesting (and updated) discussion and conclusion.

Example of literature based on parents' opinion on youth drinking and collected with qualitative methods are listed.

Østergaard, J. (2009). Learning to become an alcohol user: Adolescents taking risks and parents living with uncertainty. Addiction Research and Theory, 17(1), 30–53.

Rolando, S., Beccaria, F., Petrilli, E., & Prina, F. (2014). Adults’ views of young people's drinking in Italy: An explorative qualitative research. Drugs: education, prevention and policy21(5), 388-397.

(BTW the latter study was suggested in the previous review, but was apparently misunderstood and confused with another publication (Rolando and Katainen) which was actually on alcoholism and it is not correctely quoted on P. 8 line 501 (it is not that italian and finnish young people do know much about BD...) 

Following the suggestions of the reviewer the quote of "Rolando and Katainen" on P. 8 line 519 was deleted:

Regarding the cognitive factors, we found there is little knowledge about BD in both adolescents and mothers, as in the study by Jander et al. [23]…” (lines 519-520).

In addition, following the suggestions of the reviewer we have modified the text as follows:

Introduction

“Furthermore, previous research demonstrated the importance of parental influence on BD in adolescents. Currently, attitudes toward traditional alcohol-raising practices currently appear to be ambiguous and they are being called into question [21]. On the one hand, some qualitative research established that letting children drink at home with them can be protective against BD [20,21]. In this sense, a focus group study with parents conducted in Italy stated that allowing children to taste alcohol for the first time at home and under the parents’ supervision can be an effective way to establish a clear dialogue between parents and children, with the aim of improving shared rules and parental control [21]. Similarly, another qualitative study conducted with Danish parents established that instead of trying to ban adolescents drinking, they try to minimize the risk by helping them learn to “master intoxication” by controlling the amount and type of alcohol that they consume [20]. Instead, other studies established that high parental alcohol consumption levels [23,29,32], low parental supervision, non-rejection of BD and unstructured households [1,33], as well as less strict alcohol-related rules [34,35] were directly related to the alcohol consumption and BD of adolescents. In Spain, parental permissiveness regarding their children’s alcohol consumption is high: 48.3% of adolescents are allowed to drink alcohol by their parents [8]. As we found opposite finding in the different studies, we highlight the importance of continuing with the research on parental views of alcohol consumption and BD in their children, which could guide the development of effective health programmes to address BD [19].”. (lines 86-103).

Discussion

“Culture plays the main role in shaping the way people learn to drink and establish their alcohol consumption patterns, as well as their problematic alcohol-related behaviours [21].” (lines 478-480).

“A previous study with parents claimed that adults’ opinions regarding to the alcohol consumption of adolescents seem to some extent to be affected by an excessive alarm spread by media messages [21].” (lines 485-487).

“In this line, in a previous study with focus groups, authors claimed that both girls and boys are reluctant to associate women with a variety of alcohol-related activities, including being intoxicated. Besides, this sexual stereotyping appears present at a relatively young age and continues through adolescent experimentation with alcohol [20].” (lines 495-498).

“In contrast, Rolando et al., [21] claimed that adolescents were active actors whose drinking, like adults’, is based on a conscious and rational choice, even they believe adolescents are more aware of this risk than adults.” (lines 533-535).

“However, Rolando et al. [21], state that the traditional alcohol-raising practices based on applying strict rules, currently are being called into question. In their findings it was shown that parents are more and more convinced that it is better to socialize their children to alcohol at home [20,21], which could help children to learn to “master intoxication” by controlling the amount and type of alcohol that they consume [20].” (lines 573-577).

2) Conclusions

Conclusions are not very specific or new (it well known that prevention should be gandered or that it should aim to improve self-exteem). In my opinion, conclusions and dicussion should better highlight what the approach based on the I-Change Model add to the previous knowledge. 

According to the previous advice from the reviewer, the conclusion section was modified:

This study reports the perceptions of adolescents and mothers about determinants of BD in Spanish adolescents based on the I-Change Model, and how families manage their children's drinking behaviour. The findings can help Nurse Practitioners and healthcare professionals who work in Primary Care and School Health by developing health communication programmes to reduce BD. Our findings highlight that both adolescents and parents reported on gender differences, since the awareness factors, social influence of family and the people around them, attitude towards consumption, and self-efficacy between others could vary according to gender. For example, boys list more advantages of BD, did not perceive any danger in BD, recognise less short-term consequences of BD, and reported less parental control than girls, among other features. In addition, it was highlight that self-esteem and self-efficacy as well as the importance of increasing the risk perception, which could be reach, for example, showing situations where adolescents feel identified, which could attract their attention, improve the awareness and favour pre-motivational and motivational components of adolescents to decrease BD. Furthermore, our findings highlight the importance of increasing parents’ awareness factors, communication skills, adequate family influence, parental supervision and control of adolescents, and thus reduce BD. In sum, we believe that all these results could help to design health communication advice.” (lines 619-634).

p.10 lines 602-604:"We should also focus on promoting self-esteem and self-efficacy as well as showing real cases in order to shock adolescents into awareness, and favour pre-motivational and motivational components of BD in adolescents" --> I would recommend to avoid to give such a suggestion, since it is well settled that the "fear-appeal approch" does not work in prevention!!

We realize that we did not explain well in such paragraph. When we refer to showing real situation, we do not refer to "fear-appeal approch". What we highlighted is that, according to adolescents’ perceptions, in health promotion, real life should be reflected, which we could be translate in the importance of increasing the risk perception. The sentence was modified as follows:

In addition, it was highlighted that self-esteem and self-efficacy as well as the importance of increasing the risk perception. This could be reach, for example, showing situations where adolescents feel identified, which could attract their attention, improve the awareness and favour pre-motivational and motivational components of adolescents to decrease BD. Furthermore, our findings highlight the importance of increasing parents’ awareness factors, communication skills, adequate family influence, parental supervision and control of adolescents, and thus reduce BD. In sum, we believe that all these results could help to design health communication advice.” (lines 627-634).

If you want to give practical advice to prevention, you should also deal with prevention litarature. Otherwise, keep the discussion and conclusion more focused on your theretical approach and what it adds to knowledge about the topic under studies and how it can be used to improve interventions.

We modified the conclusion focus on our theoretical approach and what our study adds to knowledge about the topic and how it can be used to improve the health interventions, in the same paragraph that is presented above.

General observation: here and there results are defined as objective, eg in the conclusion: "This study reports the determinants of BD in Spanish adolescents". This can sound misleading and you you should pay attention to clarify that you have investigated perceptions and opinions about determinants, rather than determinants.

 According to the previous advice from the reviewer, we have modified the text as follows:

Abstract

The aim of this study was to explore the perceptions about determinants of binge drinking in Spanish adolescents from the perspective of adolescents and parents.” (lines 16-18).

Introduction

The objective of this paper is to explore the perceptions about determinants of BD in Spanish adolescents from the perspective of adolescents and parents, considering gender differences and whether they are engaging in BD.” (lines 114-116).

Discussion

The main purpose was to explore the perceptions about determinants of BD in Spanish adolescents from the perspective of adolescents and parents, taking into account gender differences and whether they are engaging in BD.” (lines 469-471).

Conclusion

This study reports the perceptions of adolescents and mothers about determinants of BD in Spanish adolescents based on the I-Change Model, and how families manage their children's drinking behaviour.” (lines 619-621).

This manuscript is a resubmission of an earlier submission. The following is a list of the peer review reports and author responses from that submission.

Round 1

Reviewer 1 Report

Whilst I think this is a potentially interesting study the current version asks too much of the reader.  My main concern is the presentation of the results.   Although the quotes are mapped to the statements this requires the reader to move between a mass of text and the main body of the paper.   Each quote should be mapped to the main body of the text.  I am afraid I found the manuscript unreadable.   In my opinion this requires a significant re-write and restructuring

I applaud the presentation of verbatim quotes however it is unclear whether this is the full extent of all the data that was collected.  It would be helpful to know how much data was collected in terms of word lengths of transcripts etc.  Currently it is not clear why these statements have been selected.

The strengths and limitations section is generally good but you have paid little attention to group dynamics- especially in such large groups.  How did you ensure the discussion was not dominated by particularly assertive individuals?

What about power?  In the mixed focus groups there is a high likelihood that males could dominate (was this the case?) Did you take any steps to prevent this?  The other issue is individuals making comments to conform to group norms.  All of these are possibilities that should be acknowledged even there is little you could do prevent them.  It is important to acknowledge them.

What about your own role in the research?  This will impact on how you chose to present and interpret the data.  Did you take steps such as reflections and checking to lessen this possibility?   Again you cannot eliminate this but you should take steps to minimise this or at least acknowledge the possibility.

Reviewer 2 Report

The research design is really interesting and deserve attention beacause of scarsity of qualitative studies with large data sets, which make possible to make multi comparisons (Young/parents, M/F, BD/not BD, age).

However, a number of weaknesses undermine the potential of the analysis and the reasoning remains quite superficial to be a qualitative in depth analysis. Maybe this is also due to the fact that the authors want to analyse too many aspects/targets in only one paper.

My main concerns are listed following.

Introduction

Overall: too superficial in explaining the phenomenon

lines 36-41: 1) the definition of BD is taken for granted, the authors should acknowledge that there is no a consensus definition (see eg Donovan 2009 and Herring, Berridge, and Thom 2008), the use of this concept has been questioned and limits have been shown. (A few examples: differences in time-spams; BD can be either intoxication oriented or not - but sometimes in the analysis it is used as synonymous of drunkenness; differences between drinking cultures make it difficult to use it in different countries…). See for instance Gmel, Rehm, and Kuntsche 2003, Thickett et al. 2013, Beccaria et al. 2015, Katainen & Rolando 2014

2) BD in Spain is not well explained: ref. to 2001 data is too dated, furthermore this data is below the EU mean and this should be acknowledged. Indeed, BD in botellon might be different from UK BD, as the authors state at p. 59-60, however, in the rest of the paper the data are scarcely contextualized and peculiarities of Spanish youth drinking are not discussed. The specific Spanish (youth) drinking culture is not adequately described.

3) Line 37. Is youth BD growing in EU? More recent studies show the opposite in many western countries, including EU see e.g. Pape et al. 2018

Lines 53-73: Evidences are all about determinants of BD, risk and protective factors as emerged from studies different for methodological approach (epidemiological, longitudinal, on personality…) and from countries with a different drinking culture. Since the paper is based on FGs and young people/parents’ perspectives I would have expected to find references to other studies based on FGs with young people and parents. Alcohol literature provides several studies about both (see eg. Rolando et al. 2014). If not here, they should be acknowledged at least in the discussion, so to improve the salience of the results. See also comments to the analysis.

Information from literature are simply listed in sequence of quite simple/short statements without a critical perspective or a clear thread of discourse. E.g. lines 68-73 give scuttered data about gender differences, which are actually such a complicated issue that dismissing it in 5 lines sounds a bit naïve.

Methods

The mean number of young participants for group is fine, but not that of parents: less of five participants per group is not proper. Please justify and list it as a limit, together with the fact that all participants were women except than one father. Results should be discussed as mothers’ point of view, rather than parents’.

About the young people groups: do the participants came from same classrooms and/or know each other? How did this potentially conditionate the discussions?

Table 2 should be moved here as description of the sample, it’s not a result!

How were the parents grouped?

Analysis

Main concern: the use of qualitative data. In my view the paper would be improved by using qualitative data (quotations) within the main text, as examples supporting what is claimed.

The FGs “nature” of data is not exploited, the natural flow of the discussion, quoted on p. 3 line 107 is not accounted. Usually in FG participants there are different levels of agreement/disagreement around different topics, and often fluctuations in the opinions based on interaction, “opinion leader”... The analysis (as well as verbatim) does not take into account dynamics/interactions, but remains quite superficial, like participants had just answered shortly to a number of questions one by one, rather than having being involved in group conversations.

The list of quotations (verbatim results) provided in Tables 3-5 are really long. I think it is not necessary to put so many quotes, while it is crucial to contextualize them in a broader and more explanatory discourse. Readers can find it quite difficult to read and check/combined the quotes in the boxes with the results. Furthermore, quotations are not related to different target groups. The aim of the authors seem to be that of providing the reader an account of the material, but this can’t be provided as raw data.  

On the other hand, a little room is left to the main text describing results, which is quite synthetic and segmented, ended up to sounds as simplistic and superficial. Complex concepts (such as patterns, self-exteem, beliefs, parental influence…) are therefore treated very briefly, as well as differences across targets – whose overall meanings are not easy to get.

Though it acknowledged that “none of adolescents were familiar with the definition of BD” p. 8 line 179, this is not accounted as a limit of the study not discussed further.

The same considerations should be done about results from parents’ groups.

Discussion

Main concern: the authors come back to literature (of different nature from qualitative studies) about determinants and factors, as this study might confirm or disconfirm them. It should actually deepen the knowledge about attitudes of Spanish adolescents and their parents about BD: what does this study add to the previous knowledge? What is the added value of having used FGs compared to other kinds of studies? In what the results differ from similar studies, carried in other countries (e.g. in UK, since most used references refer to it)?

Since so many variables are under study, many complex concepts find not a proper room to be explained, for instance the concept of self-esteem – which seems to be central also in relation to conclusions, but not well defined in relation to actual data nor to the literature.

Also differences between young people vs parents’ view (one of the strengths and originality of the study, according to the same authors) are not enough focused and addressed.

Limitations

Limits quoted in the previous text should be added (especially about parents/mothers’ group)

Strenghts: lines 410-412, this aspect seems not well exploited, at least not in term of comparison

Line 413 the Spanish context is not actually explained, so any differences with other countries are not apparent nor discussed.

Conclusions

Since the arguments are not always easy to follow, the conclusions about prevention strategies should be more explicit. e.g. in what ways should be gendered?